# TGF-β1-Mediated FDNCR1 Regulates Porcine Preadipocyte Differentiation via the TGF-β Signaling Pathway

**DOI:** 10.3390/ani10081399

**Published:** 2020-08-11

**Authors:** Zhe Zhang, Yu Meng, Fei Gao, Yue Xiao, Yi Zheng, Hao-Qi Wang, Yan Gao, Hao Jiang, Bao Yuan, Jia-Bao Zhang

**Affiliations:** 1Department of Laboratory Animals, Jilin Provincial Key Laboratory of Animal Model, Jilin University, Changchun, Jilin 130062, China; zhangz17@mails.jlu.edu.cn (Z.Z.); mengyu9917@mails.jlu.edu.cn (Y.M.); gaofei1986@jlu.edu.cn (F.G.); xiaoyue19@mails.jlu.edu.cn (Y.X.); yizheng18@mails.jlu.edu.cn (Y.Z.); hqwang1997@163.com (H.-Q.W.); gyan08@163.com (Y.G.); jhhaojiang@jlu.edu.cn (H.J.); 2College of Animal Science and Technology, Northwest A&F University, Yangling, Shanxi 712100, China

**Keywords:** TGF-β1, FDNCR1, porcine preadipocytes, differentiation, TGF-β signaling pathway

## Abstract

**Simple Summary:**

Fat differentiation affects lipid deposition and is a complex metabolic process. It has been previously reported that multiple transcription factors regulate adipocyte formation. With the in-depth study of epigenetics, in recent years it has been reported that long noncoding RNA (lncRNA) can effectively affect the formation of lipid droplets and thus regulate fat deposition. lncRNA can regulate cell function through a variety of mechanisms, the most studied is the mechanism of action of lncRNA as a miRNA molecular sponge. The purpose of this article is to explore the role of transforming growth factor-beta (TGF-β1) mediated lncRNA in the formation of porcine adipocytes, from the perspective of lncRNA to reveal the effect of TGF-β1 on the differentiation of porcine adipocytes, and provide a new way to improve the quality of pork.

**Abstract:**

Adipocyte differentiation and lipid metabolism have important regulatory effects on the quality of meat from livestock. A variety of transcription factors regulate preadipocyte differentiation. Several studies have revealed that transforming growth factor-beta (TGF-β1) may play a key role in epithelial–mesenchymal transition (EMT); however, little is known about the effects of TGF-β1 treatment on porcine preadipocytes. To explore the role of TGF-β1 in porcine adipocyte differentiation, porcine preadipocytes were treated with 10 ng/mL TGF-β1, and two libraries were constructed for RNA-seq. We chose an abundant and differentially expressed long noncoding RNA (lncRNA), which we named fat deposition-associated long noncoding RNA1 (FDNCR1), for further study. RT-qPCR was used to detect mRNA levels of genes related to adipocyte differentiation. Triglyceride assay kits were used to detect lipid droplet deposition. TGF-β1 significantly suppressed porcine preadipocyte differentiation. We identified 8158 lncRNAs in total and 39 differentially expressed lncRNAs. After transfection with FDNCR1 siRNA, the mRNA expression of aP2, C/EBPα, and PPARγ and triglyceride levels significantly increased. Transfection with FDNCR1 siRNA significantly decreased protein levels of *p*-Smad2/Smad2 and *p*-Smad3/Smad3. These results demonstrate that FDNCR1 suppresses porcine preadipocyte differentiation via the TGF-β signaling pathway.

## 1. Introduction

Requirements for pork quality have increased with improvements in living standards. Pork quality is affected by fiber diameter, color, pH, intramuscular fat, creatine levels, and fatty acid composition [1]. Among these factors, intramuscular fat content has close relationships with the juiciness, tenderness, flavor, color, and other characteristics of pork and is thus one of the key factors affecting the sensory quality of pork [2,3]. Research has shown that adipose tissue is directly related to the yield and quality of meat. Adipose tissue in animals is mainly distributed subcutaneously on abdominal tissues or on surfaces of internal organs. Two types of fat are deposited in muscle: intermuscular fat and intramuscular fat [4]. Intermuscular fat is deposited in the form of lipid droplets within muscle fibers. In high-quality pork, intramuscular fat cells accumulate large numbers of fat droplets that result in visible marbling [5]. The ideal intramuscular fat content of pork is generally considered to be 2.2–3.4%. When the intramuscular fat content is less than 2%, the texture and taste of the meat are relatively poor. Therefore, studying factors that affect fat deposition will help improve the quality of pork.

Transforming growth factor-beta (TGF-β) signaling influences a wide variety of processes, such as autophagy, differentiation, and apoptosis [6]. TGF-β consists of TGF-β1, TGF-β2, and TGF-β3 [7]. Activated TGF-β ligands initiate signal transduction by binding to TGFBR1 and TGFBR2; initially, ligands bind to TGFBR2, which recruits and then phosphorylates TGFBR1. Phosphorylated TGFBR1 subsequently facilitates the phosphorylation of R-Smads (Smad 2 and Smad 3). Following phosphorylation, R-Smads form complexes with Smad 4 and accumulate in the nucleus to transmit cell signals [7,8]. Smads 2 and 3 have been identified as activators of the TGF-β pathway, whereas Smads 6 and 7 have been identified as inhibitors of the TGF-β pathway [9,10]. TGF-β induces epithelial–mesenchymal transition (EMT) in epithelial cells and can regulate the expression of key EMT factors in vitro [11]. TGF-β signaling regulates myofibroblastic differentiation and the production of extracellular matrix (ECM) components by myofibroblasts [12]. Previous studies have shown that TGF-β1 inhibits adipocyte differentiation in obese mice [13]. However, it is unclear whether TGF-β1 has the same effect on porcine preadipocytes.

Long noncoding RNAs (lncRNAs) are a class of noncoding RNAs that are longer than 200 nucleotides (nt) [14], generally have two to four exons [15], and are less conserved than mRNAs. Some studies have shown that approximately 90–95% of transcripts are noncoding RNAs [16], and lncRNAs are thought to serve four functions, which include acting as scaffolds, guides, decoys, and signals [17]. LncRNAs have been studied in a variety of mammals and have been found to play essential roles in diverse biological processes, such as transcriptional regulation [18], genomic imprinting, metabolism, and epigenetic regulation [19]. LncRNAs have also been shown to play vital roles in tumor initiation, progression, and metastasis [20]. Although many studies have described the functions of lncRNAs, little is known about their regulatory roles in porcine adipocytes.

To study the molecular mechanism of TGF-β1 in porcine preadipocyte differentiation, we performed lncRNA sequencing (RNA-seq) of TGF-β1-treated porcine preadipocytes, and we identified a differentially expressed lncRNA, named fat deposition-associated long noncoding RNA1 (FDNCR1). We explored whether FDNCR1, a competitive endogenous RNA (ceRNA) of miR-204, regulates preadipocyte differentiation in porcine preadipocytes. Our findings will expand the understanding of the lncRNA regulation of meat quality and will help to improve porcine genetics and breeding in China.

## 2. Materials and Methods

### 2.1. Ethics Statement

The experiment was carried out in strict accordance with the guidelines of the Care and Use of Laboratory Animals of Jilin University. In addition, the whole experimental process was approved by the Institutional Animal Care and Use Committee of Jilin University (license number: 201809042).

### 2.2. Cell Culture and Differentiation

Porcine preadipocytes were treated with 10 ng/mL recombinant human TGF-β1 (PeproTech, USA) for 48 h or with 10 mM citric acid (Solarbio, China) as a negative control (NC) when the cell density reached approximately 70–80%. After 48 h of treatment and culture at 37 °C with 5% CO_2_, cells were harvested for RNA isolation. Porcine preadipocyte culture and differentiation were performed as described in a previous study [21].

### 2.3. RNA Isolation, Library Construction, and RNA-Seq

Total RNA was isolated from porcine preadipocytes treated with 10 ng/mL TGF-β or 10 mM citric acid for 48 h. Six samples of total RNA were extracted using the TRIzol method (Roche, Belmont, CA, USA) according to standard protocols. The TGF-β1 group and the NC group each included three biological replicates. A Ribo-Zero Gold rRNA Removal Kit was used to remove rRNA from RNA following the manufacturer’s instructions. An AMPure XP system was used to purify library fragments and isolate cDNA fragments that were 150–200 bp in length. An Agilent 2100, a Qubit 2.0 and RT-PCR were used to assess the cDNA library quality and quantity. The insert size was assessed on a machine and compared to the expected size. An Illumina HiSeqXten platform was used to sequence constructed libraries at BioMarker Technologies (Beijing, China) following the manufacturer’s recommended protocol.

### 2.4. Transcriptome Assembly and Data Analysis

After removing low-quality reads, adapters, and poly-N sequences, we obtained processed data from the raw data and calculated the GC content, Q20, and Q30 of the clean data. Susscrofa 10.2 was downloaded from the Ensembl genome browser and was used as the reference genome for isoform expression level quantification, splice junction searching, and alignment. To explore expression levels of all porcine genes and transcripts in the two libraries, StringTie1.3.1, an algorithm based on optimization theory, was used to calculate the mapped values of the two libraries.

### 2.5. Pipeline to Identify LncRNAs

Assembled novel transcripts were screened from two libraries to identify lncRNAs via several steps. The fragments per kilobase per million mapped reads (FPKM) value of each transcript was calculated using StringTie 1.3.1, and transcripts with values larger than 0.1 were considered to be expressed. We screened transcripts longer than 200 nt with two exons as lncRNA candidates. The Pfam database [22], CNCI(Coding-Non-Coding Index) [23], CPAT(Coding Potential Assessment Tool) [24], and CPC (Coding Potential Calculator)Tools [25] were used to ensure that each transcript was a lncRNA with protein-coding ability. Finally, we used Cuffcompare to place lncRNA transcripts into four categories, which included antisense lncRNA, sense lncRNA, lincRNA, and intronic lncRNA. The Bioconductor package (limma, version 3.4.0) and R software (version 3.4.0) with default parameters were used to analyze differentially expressed lncRNAs and mRNAs [26]. We used the Benjamini and Hochberg method to correct for multiple testing. The threshold for significantly different expression was set as a false discovery rate (FDR) < 0.1 and a fold change (FC) > 1. In the process to detect differentially expressed genes (DEGs), the FC was defined as the ratio of expression levels in two samples (groups), and the FDR was obtained by correcting the *p* value for each significant DEG.

### 2.6. Enrichment Analysis

We searched for differentially expressed lncRNAs [27] based on their expression levels and predicted their functional roles. We used the Gene Ontology Enrichment Analysis Software Toolkit (GOEAST) and the Database for Annotation, Visualization, and Integrated Discovery (DAVID) to predict the functions of abnormally expressed mRNAs [28]. GO (Gene Ontology) analysis showed that numerous host genes had strong relationships with molecular functions, cellular components, and biological processes. The topGOR software package was used to analyze DEGs. The Kyoto Encyclopedia of Genes and Genomes (KEGG) database is the main public database for pathway analysis; we next performed KEGG pathway enrichment analysis of DEGs or lncRNA target genes using KEGG Orthology-Based Annotation System (KOBAS) software [29].

### 2.7. RT-qPCR and Western Blot Analysis

We used RT-qPCR to validate the RNA-seq results. RT-qPCR was performed on a Mastercycler ep Realplex2 system with 2× SuperRealPreMix Plus (SYBR Green) and a LncRNA qPCR Detection Kit following the manufacturer’s instructions (Appendix A). Porcine preadipocytes were separated by 12% SDS-PAGE. Next, proteins were transferred to PVDF membranes (Millipore, Burlington, MA, USA). Membranes were incubated with primary antibodies at 4 °C and then incubated with a secondary antibody. Proteins were visualized using a chemiluminescent substrate (Tanon, China).

### 2.8. Oil Red O Staining and Triglyceride Detection

An Oil Red O Kit (Solarbio, USA) was used to stain porcine preadipocytes according to the manufacturer’s instructions. The morphology of lipid droplets after oil red O staining was observed under a microscope. A triglyceride kit (Applygen, China) was used to measure preadipocyte lipidosis following the manufacturer’s recommended protocol.

### 2.9. FISH

FITC-labeled FDNCR1 probes were obtained from RiboBio (Guangzhou, China). FISH was performed using a FISH kit (RiboBio) following the manufacturer’s instructions.

### 2.10. Statistical Analysis

Data are shown as the means ± standard error of the means (SEMs). Data were analyzed using SPSS 11.5 software and GraphPad Prism 5.0 software. Independent-samples *t*-tests were used to analyze data from two samples. Multiple comparisons were performed with one-way ANOVA, and a value of *p* < 0.05 was considered significant.

## 3. Results

### 3.1. Effects of TGF-β1 on Porcine Preadipocyte Differentiation

After treatment with 10 ng/mL TGF-β1 for 48 h, treated porcine preadipocytes showed significantly decreased levels of PPARγ, aP2, and C/EBPα mRNA (Figure 1A). Compared to the control (untreated) condition, treatment with 10 ng/mL TGF-β1 for 8 days significantly decreased lipid accumulation (Figure 1B,C), as indicated by staining with an Oil Red O Kit (Solarbio, Beijing, China) following the manufacturer’s instructions. After treatment with 10 ng/mL TGF-β1 for 48 h, treated porcine preadipocytes showed significantly decreased levels of PPARγ protein (Figure 1D,E). These results suggest that TGF-β1 inhibits porcine preadipocyte differentiation and lipid production.

### 3.2. Overview of RNA-Seq

To investigate the lncRNA expression profiles in porcine preadipocytes, two RNA-seq libraries were constructed from cultured porcine preadipocytes with or without TGF-β1 treatment. There were three biological replicates each in the control and treatment groups. RNA-Seq was used to sequence cDNA libraries. A total of 279,969,676 and 249,341,424 raw reads were obtained from the TGF-β1 and control libraries, respectively. Processed sequencing data were aligned to the reference genome (Susscrofa 10.2), and 87.69–88.08% of the data were successfully mapped. The Pfam database, Coding-Non-Coding Index (CNCI), Coding-Potential Assessment Tool (CPAT), and Cardiovascular Proteomics Center (CPC) Tools were used for intersection analysis to identify 8158 lncRNAs (Figure 2A), which included 4952 long intergenic noncoding RNAs (lincRNAs; 60.7%), 1367 antisense lncRNAs (16.8%), 1173 intronic lncRNAs (14.4%), and 666 sense lncRNAs (8.2%) (Figure 2B). A total of 39 lncRNAs and 76 mRNAs were differentially expressed in porcine adipocytes treated with 10 ng/mL TGF-β1 compared to control porcine adipocytes (Appendix A). Among these differentially expressed lncRNAs, 28 were upregulated, and 11 were downregulated. We identified 212,064 protein-coding transcripts, and the lengths of mRNAs and lncRNAs were mainly 200–800 bp (Figure 2C). Most mRNA transcripts had 1 exon, while lncRNA transcripts had 2–4 exons (Figure 2D); 69.82% of lncRNAs had two exons. The distribution of lncRNAs and mRNAs on porcine chromosomes is shown in Figure 2E. Heat maps were constructed to show differentially expressed lncRNAs in two different libraries (Figure 2F).

### 3.3. Identification of FDNCR1

According to the sequencing results, FDNCR1 was located on chromosome 16 in pigs (Figure 3A). In addition, we used fluorescence in situ hybridization (FISH) to determine the position of FDNCR1 in cells (Figure 3B). FISH results showed that FDNCR1 was mainly located in the cytoplasm. These results suggest that FDNCR1 may be involved in the differentiation of porcine precursors by acting as a molecular sponge. We summarized the miRNAs that target FDNCR1 (Appendix A). In addition, we detected the expression of FDNCR1 in different tissues and at different developmental stages. FDNCR1 expression decreased from day 0 to day 6 of preadipocyte differentiation (Figure 3C). FDNCR1 was more abundant in muscle and lung tissue than in other tissues (Figure 3D).

### 3.4. Knockdown of FDNCR1 Promotes Preadipocyte Differentiation

The efficiency of FDNCR1 knockdown was evaluated by RT-qPCR analysis (Figure 4A). The knockdown efficiency of FDNCR1 siRNA-2 was the best, so we used FDNCR1 siRNA-2 in a follow-up test. The results showed that compared with the control treatment, siRNA transfection significantly increased mRNA levels of PPARγ, aP2, and C/EBPα (Figure 4B). FDNCR1 siRNA also significantly increased the formation of lipid droplets in preadipocytes (Figure 4C) and the triglyceride content (Figure 4D). These results indicated that the knockdown of FDNCR1 promoted the porcine preadipocyte differentiation.

### 3.5. Knockdown of FDNCR1 Promotes Preadipocyte Differentiation through the TGF-β Signaling Pathway

After transfecting preadipocytes with FDNCR1 siRNA, we used RT-qPCR to detect the expression of miR-204 and TGFBR1 to examine the regulatory effect of FDNCR1. FDNCR1 siRNA significantly increased the expression of miR-204 but decreased the expression of TGFBR1 in preadipocytes; however, cotransfection with FDNCR1 siRNA and a miR-204 inhibitor restored miR-204 and TGFBR1 expression (Figure 5A,B). Transfection with FDNCR1 siRNA significantly decreased protein levels of *p*-Smad2/Smad2 and *p*-Smad3/Smad3, but cotransfection with a miR-204 inhibitor restored these protein levels (Figure 5C,D). Our results suggest that FDNCR1 sponges miR-204 to regulate preadipocyte differentiation by activating the TGF-β signaling pathway (Figure 6).

## 4. Discussion

TGF-β1, an important member of the TGF-β superfamily, plays critical roles in many processes [30,31]. Tsurutani et al. showed that TGF-β1 inhibits adipocyte differentiation in obese mice [13]. However, it is unclear whether TGF-β1 has the same effect on porcine preadipocytes. This interest motivates us to explore the effect of TGF-β1 on porcine preadipocytes differentiation. In one study, treatment with 10 ng/mL human recombinant TGF-β1 for 24 h significantly decreased bovine granulosa cell proliferation, and two small RNA libraries were constructed for high-throughput sequencing [32]. Hills et al. (2018) cultured HK2 and hPTEC cells with or without 10 ng/mL TGF-β1 for 7 days to examine the expression of CX26 and CX43 [33]. Therefore, we chose 10 ng/mL TGF-β1 as our final concentration. In our study, treatment with TGF-β1 significantly increased apoptosis in porcine preadipocytes. Additionally, the mRNA levels of PPARγ, aP2, and C/EBPα were significantly decreased and lipid accumulation was significantly decreased in porcine preadipocytes treated with 10 ng/mL TGF-β1 in our study. Research has shown that PPARγ is a key regulator of adipocyte differentiation, and C/EBPα can bind to the promoter of PPARγ and regulate the differentiation of adipocytes. Our study is consistent with a report indicating that TGFβ1 inhibits adipocyte differentiation [9,13].

Fat deposition is a complex metabolic process. Previous studies have shown that inhibiting adipocyte differentiation and inducing adipocyte apoptosis can decrease body fat deposition [34]. In addition, studies have shown that lncRNAs participate in a wide range of biological processes; however, studies on the relationship between lncRNAs and fat deposition are limited. Sun et al. found that lncRNA LncIMF4 affects porcine intramuscular preadipocyte differentiation [35]. Zhang et al. showed that lncRNA RP11-142A22.4 regulates preadipocyte differentiation and acts as a sponge of miR-587 to modulate Wnt5β expression [36]. Cai et al. revealed that lncRNA BADLNCR1 regulates adipogenesis by repressing GLRX5 expression in bovine preadipocytes [37]. In this study, we validated 8158 differentially expressed lncRNA transcripts and 212,064 differentially expressed mRNA transcripts in preadipocytes treated with TGF-β1 compared with untreated preadipocytes using the Illumina HiSeqXten platform. We next investigated whether these differentially expressed lncRNAs regulated porcine preadipocyte differentiation.

Non-coding RNA is the current research focus. As an important member of noncoding RNA, lncRNA plays an important role in the field of life sciences and affects a variety of processes, such as cell proliferation [38], differentiation [39], apoptosis [40], and other processes regulating gene expression [41]. Moreover, previous studies have revealed that lncRNAs play key regulatory roles at both the transcriptional [42] and posttranscriptional levels [43]. LncRNAs exhibit complex functions through multiple mechanisms, including mRNA degradation regulation, gene imprinting, splicing regulation, chromatin remodeling, and translation regulation. Although numerous studies have reported that lncRNAs are widely expressed in humans, mice, and rats, their expression in other mammals, especially pigs, remains unknown. In our study, we detected the expression of FDNCR1 in different tissues and at different developmental stages. FDNCR1 expression decreased from day 0 to day 6 of preadipocyte differentiation. This result indicates that FDNCR1 may inhibit preadipocyte differentiation. FDNCR1 was more abundant in muscle and liver tissue than in other tissues. The fat content in muscle plays a vital role in pork [44]. Proper fat content in pork will improve the quality of pork. The fat content in the liver will affect human health, too high fat content will form fatty liver, which will seriously damage human health [45]. Therefore, the high expression of FDNCR1 in muscle and liver tissue has an important regulatory effect on human health and pork quality. The transfection of cells with FDNCR1 siRNA significantly increased the formation of lipid droplets and the triglyceride content.

Numerous previous studies have shown that miRNAs can cause gene silencing by binding to mRNA. Recently, a new potential function of lncRNAs was found: competitively binding miRNAs and reversing the inhibitory effects of miRNA on mRNA, thereby regulating biological processes in cells. This phenomenon is known as the ceRNA hypothesis [46]. LncRNAs play important roles in various diseases by competitively binding miRNAs. For example, the lncRNA SMAD5-AS1, a sponge of miR-135b-5p, inhibits cell proliferation via the Wnt/β-catenin pathway in lymphoma [47]. The lncRNA MIAT sponges miR-22-3p to upregulate DAPK2 in diabetic cardiomyopathy [48]. However, relatively few lncRNAs have been identified that affect fat deposition through porcine preadipocyte differentiation by competitively binding miRNAs. In this study, after transfecting cells with FDNCR1 siRNA, mRNA levels of miR-204 significantly increased. Our previous research showed that miR-204 and miR-181 regulate porcine preadipocyte differentiation by targeting TGFBR1 [21,49]. We speculate that lncRNAs regulate TGFBR1 expression by competitively binding to miR-204. Notably, transfecting cells with FDNCR1 siRNA significantly decreased mRNA levels of TGFBR1 and protein levels of *p*-Smad2/Smad2 and *p*-Smad3/Smad3.

## 5. Conclusions

In our study, we revealed that FDNCR1 competitively binds to miR-204 to regulate the expression of TGFBR1, thereby regulating porcine preadipocyte differentiation. Our research will provide valuable information for improving traits of porcine meat quality. Our study also provides new ideas for further study and identified a novel target and a new strategy to control fat deposition.

## Figures and Tables

**Figure 1 animals-10-01399-f001:**
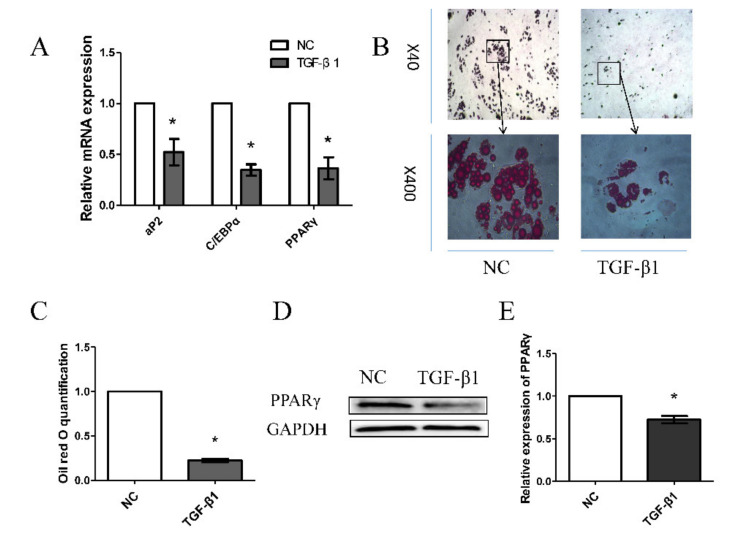
Morphology of preadipocytes treated with 10 ng/mL transforming growth factor-beta 1 (TGF-β1). (**A**) Relative mRNA levels with or without treatment with 10 ng/mL TGF-β1 for 48 h. (**B**,**C**) Oil red O staining (40× and 400×) showing lipid accumulation on day 8. (**D**,**E**) Relative protein levels of PPARγ and the apoptosis rates of porcine preadipocytes. Each treatment was repeated three times. Quantitative data are presented as the mean ± standard deviation (SD). “*” indicates significance.

**Figure 2 animals-10-01399-f002:**
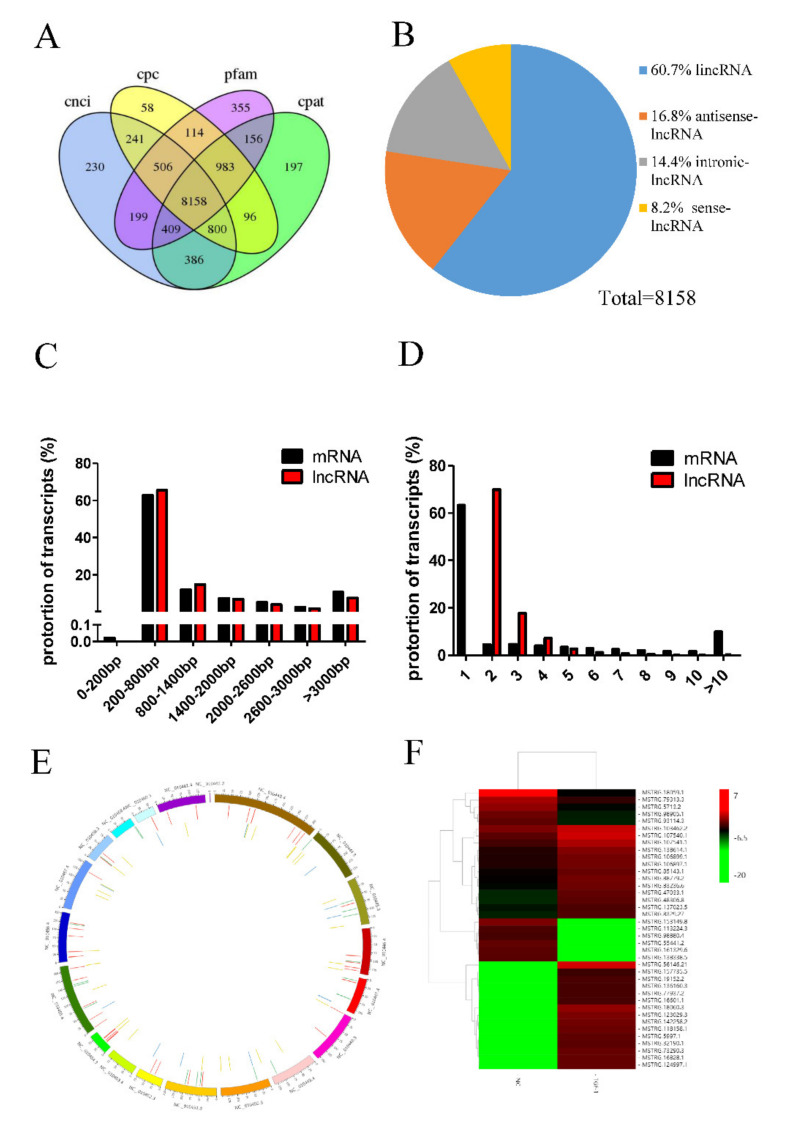
Overview of long noncoding RNA (lncRNA) sequencing data. (**A**) The CNCI(Coding-Non-Coding Index), Pfam database, CPC (Coding Potential Calculator)Tools, and CPAT (Coding Potential Assessment Tool)were used to predict the potential coding capability of lncRNAs to identify lncRNAs. (**B**) Types of predicted lncRNAs. Distribution of sequence length (**C**) and exon number (**D**) of lncRNAs and mRNAs. (**E**) The distribution of lncRNAs and mRNAs on porcine chromosomes. (**F**) Heat maps were constructed to show differentially expressed lncRNAs in two different libraries.

**Figure 3 animals-10-01399-f003:**
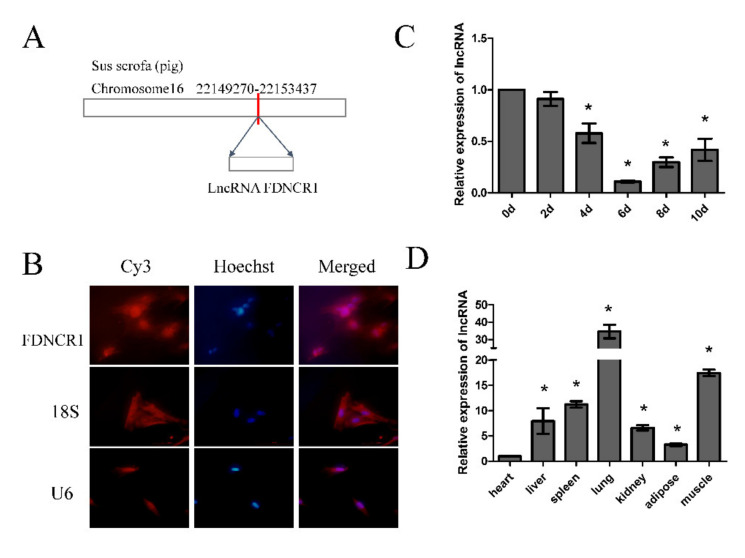
Characterization of fat deposition-associated long noncoding RNA1 (FDNCR1) in porcine preadipocytes. (**A**) Locations of FDNCR1 on pig chromosomes. (**B**) FISH detection of the lncRNA localization in preadipocytes. (**C**) Expression of FDNCR1 at different developmental stages. (**D**) Expression of FDNCR1 in different tissues. “*” indicates significance.

**Figure 4 animals-10-01399-f004:**
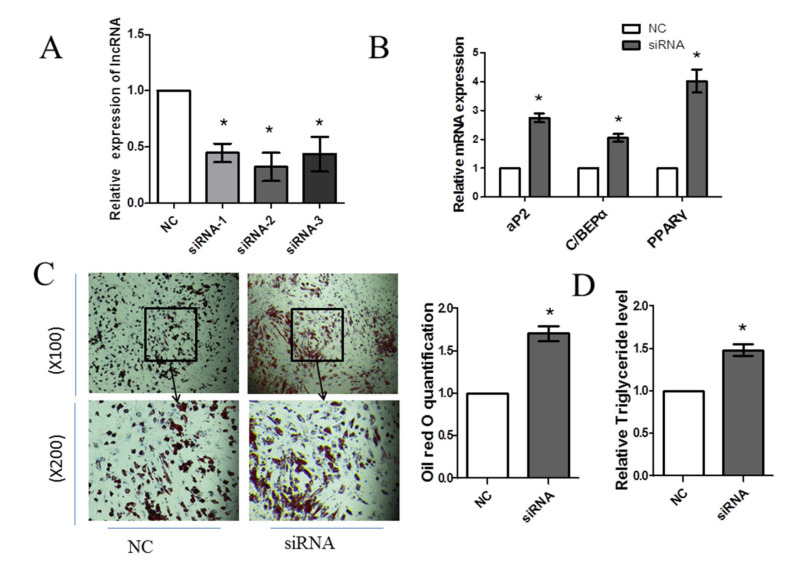
Knockdown of FDNCR1 promotes preadipocyte differentiation. (**A**) Expression of FDNCR1 after transfection with FDNCR1 siRNA. (**B**) mRNA expression of aP2, C/EBPα, and PPARγ after transfection with siRNA. (**C**) Oil red O staining (100×, 200×) showing lipid accumulation. (**D**) Levels of triglycerides after transfection with siRNA. “*” indicates significance.

**Figure 5 animals-10-01399-f005:**
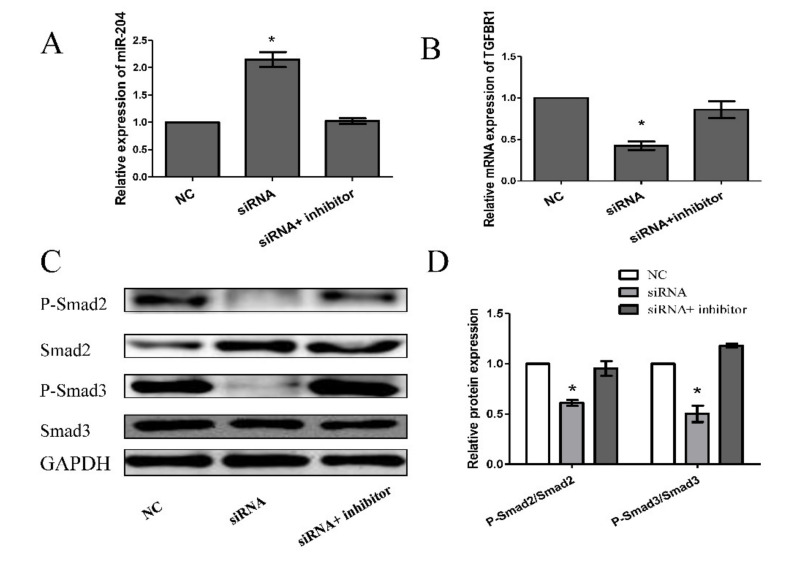
Knockdown of FDNCR1 promotes preadipocyte differentiation through the TGF-β signaling pathway. (**A**,**B**) Expression of miR-204 and TGFBR1 after transfection with FDNCR1 siRNA and/or a miR-204 inhibitor. (**C**,**D**) Protein levels of *p*-Smad2/Smad2 and *p*-Smad3/Smad3 after transfection with FDNCR1 siRNA and/or a miR-204 inhibitor. “*” indicates significance.

**Figure 6 animals-10-01399-f006:**
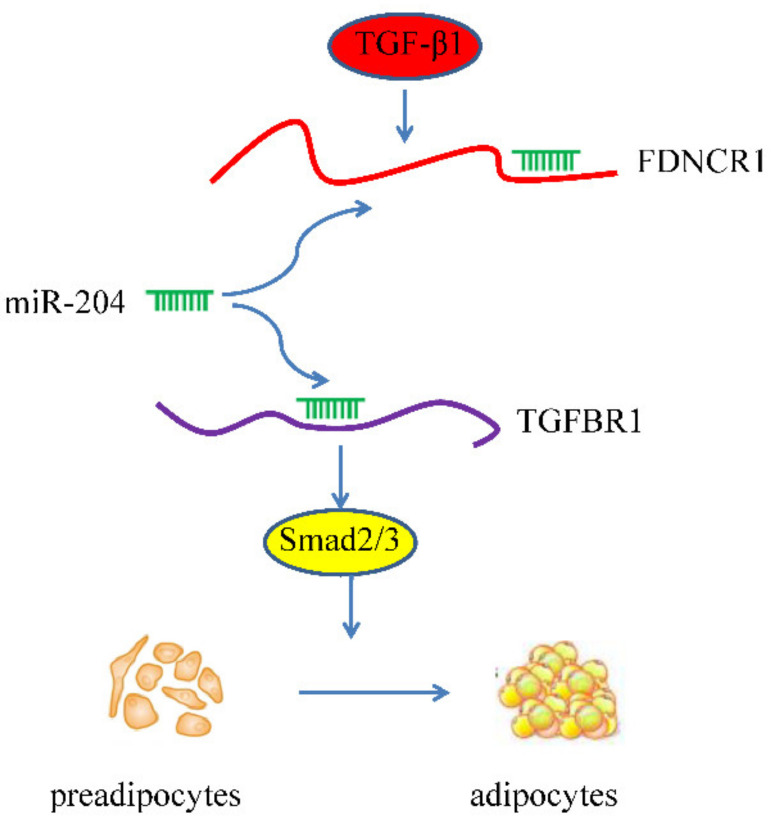
Model of the possible FDNCR1-miR-204-TGFBR1 pathway in porcine preadipocyte differentiation.

## Data Availability

Publicly available datasets were analyzed in this study. Our GEO accession number is GSE132049. These data can be found here: https://www.ncbi.nlm.nih.gov/Traces/study/?acc=PRJNA530285.

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
