# Peer review of "TGF-β1-Mediated FDNCR1 Regulates Porcine Preadipocyte Differentiation via the TGF-β Signaling Pathway"

_animals, 2020, doi:10.3390/ani10081399_

Round 1

Reviewer 1 Report

In this study, Zhe Zhang et al. showed that FDNCR1 suppresses porcine preadipocyte differentiation via the TGF-β signaling pathway. Overall this was an interesting study which proposes a novel mechanism for TGF-β in preadipocyte differentiation. Please find my specific suggestions/ comments listed below.

1) Please discuss the function of TGF-β known in preadipocyte differentiation in discussion section.

2) In the discussion, please discuss the reason why LncRNA was primarily searched in this study.

3) Please discuss the expression of FDNCR1 and its function in other tissues in discussion section.

4) In addition to FDNCR1, please discuss other LncRNAs and miRNAs known to function in adipocytes.

Author Response

Responses to Reviewers

Thank you for your detailed comments and suggestions, which we found useful as we revised our manuscript. Our responses to each of your points and suggestions are provided in the following sections.

  1. Please discuss the function of TGF-β known in preadipocyte differentiation in discussion section.

Re: Thank you for your detailed comments and suggestions.

TGF-β1, an important member of the TGF-β superfamily, plays critical roles in many processes [30, 31]. Tsurutani et al. shown that TGF-β1 inhibits adipocyte differentiation in obese mice [13]. However, it is unclear whether TGF-β1 has the same effect on porcine preadipocytes. This interest motivates us to explore the effect of TGF-β1 on porcine preadipocytes differentiation. In our study, treatment with TGF-β1 significantly increased apoptosis in porcine preadipocytes. Additionally, the mRNA levels of PPARγ, aP2 and C/EBPα were significantly decreased and lipid accumulation was significantly decreased in porcine preadipocytes treated with 10 ng/mL TGF-β1 in our study. Research has shown that PPARγ is a key regulator of adipocyte differentiation, and C/EBPα can bind to the promoter of PPARγ and regulate the differentiation of adipocytes. Our study is consistent with a report indicating that TGFβ1 inhibits adipocyte differentiation [9, 13].

We are grateful for the time and energy you expended on our behalf.

  1. In the discussion, please discuss the reason why LncRNA was primarily searched in this study.

Re: Thank you for your detailed comments and suggestions.

Non-coding RNA is the current research focus. As an important member of noncoding RNA, lncRNA plays an important role in the field of life sciences and affects a variety of processes, such as cell proliferation [38], differentiation [39], apoptosis [40], and other processes regulating gene expression [41]. Although numerous studies have reported that lncRNAs are widely expressed in humans, mice and rats, their expression in other mammals, especially pigs, remains unknown. In our study, the transfection of cells with FDNCR1 siRNA significantly increased the formation of lipid droplets and the triglyceride content.

We are grateful for the time and energy you expended on our behalf.

  1. Please discuss the expression of FDNCR1 and its function in other tissues in discussion section.

Re: Thank you for your detailed comments and suggestions.

In our study, we detected the expression of FDNCR1 in different tissues and at different developmental stages. FDNCR1 expression decreased from day 0 to day 6 of preadipocyte differentiation. This result indicates that FDNCR1 may inhibit preadipocyte differentiation. FDNCR1 was more abundant in muscle and liver tissue than in other tissues. The fat content in muscle plays a vital role in pork [44]. Proper fat content in pork will improve the quality of pork. The fat content in the liver will affect human health, too high fat content will form fatty liver, which will seriously damage human health [45]. Therefore, the high expression of FDNCR1 in muscle and liver tissue has an important regulatory effect on human health and pork quality.

We are grateful for the time and energy you expended on our behalf.

  1. In addition to FDNCR1, please discuss other LncRNAs and miRNAs known to function in adipocytes.

Re: Thank you for your detailed comments and suggestions.

In addition, studies have shown that lncRNAs participate in a wide range of biological processes; however, studies on the relationship between lncRNAs and fat deposition are limited. Sun et al. found that lncRNA LncIMF4 affects porcine intramuscular preadipocyte differentiation [35]. Zhang et al. shown that lncRNA RP11-142A22.4 regulates preadipocyte differentiation acts as a sponge of miR-587 to modulate Wnt5β expression [36]. Cai et al. revealed that lncRNA BADLNCR1 regulates adipogenesis by repressing GLRX5 expression in bovine preadipocyte [37]. In our study, we validated 8158 differentially expressed lncRNA transcripts and 212064 differentially expressed mRNA transcripts in preadipocytes treated with TGF-β1 compared with untreated preadipocytes using the Illumina HiSeqXten platform. We next investigated whether these differentially expressed lncRNAs regulated porcine preadipocyte differentiation.

We are grateful for the time and energy you expended on our behalf.

Reviewer 2 Report

this is a very important work on the molecular mechanisms and mediators of fat differentiation. The paper presents novel and relevant findings.

The authors demonstrate conclusively for the first time that TGF-ß1-mediated FDNCR1 regulates procine preadipocyte differentiation.

The TGF-ß1 signaling pathway involved is described in great detail and these findings are of utmost importance to the field.

Adipocyte differentiation has important effects on the quality of meat from livestock, thus, this is not only theoretially relevant.

The methods used are state of the art and yield new findings that shed light on the complex process of regulation.

The involvement of lncRNA and miRNA is clearly described and soundly documented by the authors.

The paper is thus relevant and interesting, the topic original, and the message clear and easy to understand.

The conclusions are sound and I really recommend the rapid publication of the manuscript without any doubt or hesitation.

The conclusions presented by the authors and fully backed by the novel findings presented in the paper.

Specific suggestions for changes:

Page 1, Line 35: the mRNA (add a space)

Author Response

Responses to Reviewers

Thank you for your detailed comments and suggestions, which we found useful as we revised our manuscript. Our responses to each of your points and suggestions are provided in the following sections.

Page 1, Line 35: the mRNA (add a space)

Re: Thank you for your detailed comments and suggestions.

We have added a space between the mRNA.

We are grateful for the time and energy you expended on our behalf.
